# Metabolite Profiling and Biological Activity Assessment of *Paeonia ostii* Anthers and Pollen Using UPLC-QTOF-MS

**DOI:** 10.3390/ijms25105462

**Published:** 2024-05-17

**Authors:** Fengfei Jiang, Mengchen Li, Linbo Huang, Hui Wang, Zhangzhen Bai, Lixin Niu, Yanlong Zhang

**Affiliations:** 1College of Landscape Architecture and Arts, Northwest A&F University, Yangling District, Xianyang 712100, China; j907609866@nwafu.edu.cn (F.J.); limengchen@nwafu.edu.cn (M.L.); hlb@nwafu.edu.cn (L.H.); 18339902483@nwafu.edu.cn (H.W.); baizhangzhen@nwafu.edu.cn (Z.B.); 2Oil Peony Engineering Technology Research Center of National Forestry Administration, Yangling District, Xianyang 712100, China

**Keywords:** *Paeonia ostii*, anther, pollen, untargeted metabolomics, biological activity

## Abstract

*Paeonia ostii* is an important economic oil and medicinal crop. Its anthers are often used to make tea in China with beneficial effects on human health. However, the metabolite profiles, as well as potential biological activities of *P. ostii* anthers and the pollen within anthers have not been systematically analyzed, which hinders the improvement of *P. ostii* utilization. With comprehensive untargeted metabolomic analysis using UPLC-QTOF-MS, we identified a total of 105 metabolites in anthers and pollen, mainly including phenylpropanoids, polyketides, organic acids, benzenoids, lipids, and organic oxygen compounds. Multivariate statistical analysis revealed the metabolite differences between anthers and pollen, with higher carbohydrates and flavonoids content in pollen and higher phenolic content in anthers. Meanwhile, both anthers and pollen extracts exhibited antioxidant activity, antibacterial activity, α-glucosidase and α-amylase inhibitory activity. In general, the anther stage of S4 showed the highest biological activity among all samples. This study illuminated the metabolites and biological activities of anthers and pollen of *P. ostii*, which supports the further utilization of them.

## 1. Introduction

Tree peony is a perennial shrub and an important economic oil crop. Tree peony seed oil has received widespread attention because its high content of unsaturated fatty acids [1]. At present, the planting area of oil tree peony has reached 130,000 hectares, and is expected to exceed 5,000,000 hectares after 5–10 years [2]. *Paeonia ostii* belongs to a kind of oil peony and is cultivated in many places in China due to its strong seed setting ability and high oil content, has become the most common variety of oil peony. Peony anthers, as a by-product of peony seed oil industrial production, have recently received attention as health-beneficial products [3]. In addition, peony anthers contain more nutrients, can be processed into tea beverage, which has broad market prospect in China. As a part of anthers, pollen has been proved to have bioactive compounds in some plants, and corresponding products have been developed [4]. However, few studies were conducted in dissecting the chemical properties of *P. ostii* anthers and pollen, as well as their bioactive compounds.

Metabolomics is a systematic technique used to understand the metabolite composition in biological systems [5]. Metabolomics can be divided into two parts: targeted and untargeted metabolomics [6]. Targeted metabolomics is mainly used for quantitative analysis of known metabolites, while untargeted metabolomics is often used for qualitative analysis of uncertain or unknown metabolites due to its wide coverage and the ability to monitor a large number of analytes simultaneously [7]. Untargeted metabolomics has been widely used in plant researches to compare metabolites among different tissues [8], across different times [9], or among tissues after different treatments [10]. At present, the metabolite composition of peony anthers and pollen is largely unknown. Therefore, in this experiment, untargeted metabolomics approach was chosen to generally analyze their metabolite profiles.

Plants are rich in secondary metabolites that mediate the interactions of plants with their environment for adaptation and defense [11]. Generally, plant secondary metabolites are classified into (a) phytoalexins, (b) hydrocarbons, (c) terpenes, (d) alkaloids, (e) phenolic compounds, (f) flavonoids, and (g) polysaccharides [12]. Secondary metabolites play a role in antioxidant [13] and antibacterial [14], making secondary metabolites unique sources of industrially important biochemical products, flavors and food additives. Furthermore, it was shown that many plant extracts can inhibit α-glucosidase and α-amylase, which further prevents diabetes [15,16]. Therefore, the discovery of novel antidiabetic drugs from plant secondary metabolites has been the focus of recent research [17].

This study compared the secondary metabolites and biological activities between anthers and pollen of *P. ostii* for the first time. The bioactivities of samples were evaluated by test their antioxidant activity, antibacterial activity, and ability to inhibit α-glucosidase and α-amylase. Secondary metabolites in all samples were analyzed and characterized with untargeted metabolomics approach. Subsequently, differentially accumulated metabolites (DAMs) among samples were identified using multivariate statistical methods. Afterwards, the correlation between bioactive compounds and bioactivities was determined, and the potential activity of secondary metabolites was predicted. This study dissected the metabolite composition and bioactivity of *P. ostii* anthers and pollen, provided a theoretical reference for their development and utilization in functional food and pharmaceuticals.

## 2. Results and Discussion

### 2.1. Morphological Characteristics

In this chapter, the phenotypic characteristics and microstructures of *P. ostii* anthers at different developmental stages were compared and analyzed, which provided a reference for harvesting of *P. ostii* anthers. As shown in Figure 1, each anther normally has four pollen sacs, which are arranged in pairs on either side of the anther (Figure 1(B1)). In the flowering stage, the inner wall cells of the anthers were elongated radially and thickened in a ribbon shape, and when the anther connectives disappeared, the microsporangia on both sides were connected (Figure 1(B2)). As the inner wall cells of the anthers thicken, the anthers dehisced and the mature pollen grains fell out (Figure 1(B3,B4)). The analysis identified the morphological differences of anther samples at different developmental stages.

### 2.2. Total Phenolic and Flavonoid Content of Tree Peony Anthers and Pollen

Phenolic and flavonoid are the most abundant metabolites in plants [18]. As shown in Figure 2, the total phenolic content (TPC) of different samples ranged from 3 mg GE/g DW (S1) to 18 mg GE/g DW (S4), there were significant differences in the TPCof different samples, and TPC accumulates dramatically from the S3 to S4 stages of the anther. The total flavonoid content (TFC) at different samples ranged from 3 mg RE/g DW (S1) to 13 mg RE/g DW (P), indicating that flavonoids are mainly distributed in pollen and less in the anther wall. The content of total flavonoid was lower than that of total phenolic, which is consistent with previous reports [19]. Comparing the TPC and TFC of *P. ostii* anthers, petals, and leaves, the anther had the lowest TPC and the highest TFC [3,20]. Therefore, the flavonoids in peony anthers, especially the pollen, have great development value.

### 2.3. Antioxidant Activity of Tree Peony Anthers and Pollen

As shown in Figure 3, there were significant differences in the antioxidant activity (DPPH, ABTS, FRAP) of samples.

DPPH^•^ (1,1-Diphenyl-2-picrylhydrazyl radical)is one of the few stable organic nitrogen radicals and its scavenging activity is one of the most commonly used methods to evaluate antioxidant activity [21]. The DPPH^•^ scavenging ability of different samples ranged from 8 mg TE/g DW (S1) to 21 mg TE/g DW (S4). DPPH^•^ scavenging activity is a hydrogen atom transfer (HAT) reaction that can acquire hydrogen atoms in some phenolic acids containing monophenol hydroxyl groups [22]. Therefore, the total phenolic content may affect the DPPH^•^ scavenging ability.

ABTS^+•^ (2,2′-Azinobis-(3-ethylbenzothiazoline-6-sulphonate) radical) scavenging reaction involves both HAT and single electron transfer (SET), which is different from DPPH^•^ scavenging ability. ABTS^+•^ scavenging reaction is also commonly used to measure the antioxidant capacity of food [23]. The ABTS^+•^ scavenging ability of stamens at different stages ranged from 10 mg TE/g DW (S1) to 41 mg TE/g DW (S4). This result is agreed with previous reports in tree peony [24].

FRAP (ferric reducing antioxidant power) is a reaction that reduces Fe^3+^ ligands by the action of antioxidants, which is different from the scavenging reactions of DPPH^•^ and ABTS^+•^. Overall, the FRAP values of all samples ranged from 3 mg TE/g DW (S1) to 18 mg TE/g DW (S4). Some researchers have shown that the FRAP value is not only dependent on the variety, but also highly influenced by geographical distribution and harvest time [25].

In order to better reflect the antioxidant activity of the samples, various detection methods are recommended in the determination process [21]. By comparing the results from different antioxidant assay experiments, we observed significant differences in antioxidant capacity among different samples. Among them, sample S4 showed better antioxidant capacity than that of other samples, which is an excellent natural antioxidant. The trends of ABTS value and FRAP value are highly consistent with the trend of the TPC of all samples, so it is reasonable to infer that phenolic compounds are the main antioxidant in peony anthers.

### 2.4. Antibacterial Activity of Tree Peony Anthers and Pollen

In this study, a total of eight common food-borne and skin-borne bacteria were used to evaluate the antibacterial activity of anthers and pollen, with cephalexin (Cef) and ampicillin (Amp) as positive controls. As shown in Table 1, it was found that there were differences in the antibacterial activity of different samples. Compared with other bacteria, all samples showed stronger antibacterial activity against *Streptococcus hemolytis-β* and *Listeria monocytogenes*. In contrast, all samples showed lower antibacterial activity against *Proteus vulgaris*. Overall, S4 and P showed stronger antibacterial activity in all samples. Studies have shown that phenolic acids and flavonoids have antibacterial effects [26]. S4 is rich in phenols and flavonoids, and pollen exhibits the highest flavonoids content, which likely enhanced their antibacterial activity.

### 2.5. Inhibition of α-Glucosidase and α-Amylase Activity of Tree Peony Anthers and Pollen

In this work, we compared the α-glucosidase and α-amylase inhibitory activity of different samples (Figure 4). It was found that there were differences in the inhibitory activity of the enzyme among different samples, and the inhibitory efficiency increased with the increase in the sample concentration.

By comparison, it was found that sample S4 (IC_50_ = 24.08 ± 2.45 μg/mL) has the strongest inhibitory activity on α-glucosidase among all samples (Figure 4A). In contrast, pollen has the weakest inhibitory activity on α-glucosidase. Some phenolic compounds, such as pentagalloylglucose and ellagic acid, have been reported to have α-glucosidase inhibitory activity [27,28]. Compared with pollen, anthers contained more phenolic compounds, which may also lead to stronger α-glucosidase inhibitory activity in anthers than that in pollen.

By comparing the α-amylase inhibitory activity of all samples, we found that S4 (IC_50_ = 130.93 ± 0.8 μg/mL) exhibited the strongest α-amylase inhibitory activity, while sample P exhibited the weakest inhibitory activity (Figure 4B). In previous studies, it was reported that differences in the content of active components among different plant tissues may cause the differences of α-amylase inhibition among samples [29]. The relationship between differentially accumulated metabolites (DAMs) and inhibition of enzyme activity is worthy of exploration in the future.

In summary, S4 has the strongest inhibitory activity on both enzymes, anthers from other three developmental stages also showed good inhibitory activity, while P showed very weak inhibitory activity on both enzymes.

### 2.6. Characterization of Metabolites by UPLC-QTOF-MS and Multivariate Statistical Analysis

#### 2.6.1. Data Characterization and Metabolite Identification

In this experiment, three biological replicates were performed for each sample, and a total of 9728 peaks were detected. Principal component analysis (PCA) with all ion peaks showed that the same group of samples could not only be clustered together but also separated from other groups of samples (Figure 5A). It is worth noting that sample S4 and P were located above the Y axis, which differed from the other groups. Likewise, in the clustering analysis, it can be found that group S1, S2, and S3 were clustered into one class, and the remaining groups were classified as one class (Appendix A). Through the correlation analysis between samples in each group, we detected high correlation between the samples in each group (Figure 5B).

The Global Natural Products Social Molecular Network (GNPS) is a cloud-based metabolite identification website. The feature based molecular network (FBMN) of this website calculates the similarity between mass spectra, and compounds with similar substructures are clustered into a molecular network [30]. The metabolites were identified using the feature based molecular network method and combined with Massbank, MoNa, ReSpect, MetaboBASE, MS-DIAL, Fiehn/Vaniya natural product library (F/V npl) and traditional Chinese medicine database (TCM) (Figure 6A). Metabolites were also identified by comparing characteristic ion fragments reported in the literature [31]. The data were deduplicated and identified a total of 105 metabolites with an error of less than 10 ppm (Appendix A). All metabolites are classified into eight categories at the superclass level (Figure 6B), including phenylpropanoids and polyketides (39%), organic acids and derivatives (11%), benzenoids (9%), lipids and lipid-like molecules (8%), organic oxygen compounds (7%), and organoheterocyclic compounds (1%). Through heatmap analysis (Figure 7), we identified metabolites differed in the content among different samples.

#### 2.6.2. Metabolite Multivariate Statistical Analysis

PCA analysis was performed on different samples based on all identified metabolites (Figure 8A), and the first principal component (PC1) and the second principal component (PC2) explained 64.2% and 24% of the total variables, respectively. The PCA results showed differences between sample groups. The quality control (QC) samples were close to the coordinate origin, which confirmed the stability of the mass spectrometry system and the accuracy of the verification data. And three replicates of each sample were clustered together, which indicating that this experiment is sufficiently reproducible. To further verify the reliability of the metabolomic data, partial least squares discriminant analysis (PLS-DA) was performed to distinguish models between different sample groups (Figure 8B). The R^2^ and Q^2^ values of this model were 0.991 and 0.987 (Appendix A), indicating the model has good predictive ability. The PLS-DA results showed that the PC1 and PC2 explained 45.1% and 43.6% of the total variables, and the larger PC1 value demonstrated differences in metabolites between anthers (S1, S2, S3, and S4) and pollen (P). The two analysis methods showed that the metabolite composition of S1, S2 and S3 are similar, while S4 and P are very different from all other samples.

#### 2.6.3. Differentially Accumulated Metabolites (DAMs) between Anther and Pollen

Through the above analysis, it was found that there were differences in metabolite composition between the anther and pollen, further analysis of DAMs screening was performed for the four comparison groups based on VIP > 1, |log2(foldchange) > 1|, and *p* < 0.05 from *t*-test (Appendix A). In general, the accumulation of metabolites differed in anthers of different developmental stages compared with pollen. To further analyze the differences in metabolites between samples, Venn diagram analysis found a total of 33 DAMs in different comparison combinations, mainly including pentagalloylglucose, methyl gallate and gallic acid hexoside (Figure 9A). Analyzing the DAMs of different comparison groups by volcano plot, we detected 59 (24 upregulated and 35 downregulated), 71 (40 and 31), 76 (45 and 31), 82 (67 and 15) metabolites that exhibited differential abundance in S1 vs. P, S2 vs. P, S3 vs. P, S4 vs. P comparison groups, respectively (Figure 9B).

Functional enrichment analysis of DAMs was performed according to the KEGG database. All DAMs are enriched to different pathways (Figure 10), mainly including “phenylpropanoid metabolism”, “starch and sucrose metabolism”, “glutathione metabolism”, “flavone and flavonol biosynthesis”, “flavonoid biosynthesis”. Among them, DAMs in all comparison groups were enriched in the “starch and sucrose metabolism” pathway. The analysis showed that anthers contain higher phenolic content, while pollen contains more flavonoids. This result is consistent with the Section 2.2 of this article, proving the accuracy of the date. Meanwhile, pollen contains more carbohydrates than anthers. Previous studies have shown that sugar metabolism in pollen promote pollen development and pollen tube growth, which is essential for pollination and fertilization [32].

### 2.7. In Silico Prediction and Correlation Analysis of Bioactive Compounds and Biological Activities

#### 2.7.1. Pearson Correlation Analysis of Bioactive Compounds and Biological Activities

Metabolites from plants have been demonstrated for biological activity, and many products have been developed [33]. As shown in Figure 11, the correlation between metabolites and biological activity was explored by Pearson correlation analysis. The result shows that most of the metabolites were correlated with biological activity (Appendix A).

It was found that there were significant correlations between TPC and DPPH (r = 0.84, *p* ≤ 0.05), ABTS (r = 0.57, *p* ≤ 0.05), and FRAP (r = 0.6, *p* ≤ 0.05). Phenolic compounds are considered to be an important natural antioxidant, as previously reported [34]. In the experiment, it was also found that the samples with high total phenolic content have stronger antioxidant capacity. Further, 68, 65 and 64 metabolites were correlated with ABTS, DPPH and FRAP activities, respectively. Interestingly, gallic acid has a strong correlation with the activities of DPPH (r = 0.61, *p* ≤ 0.05), ABTS (r = 0.95, *p* ≤ 0.05), FRAP (r = 0.93, *p* ≤ 0.05). In previous studies, it was found that gallic acid and its derivatives are a group of polyphenols, due to their phenolic hydroxyl groups that give them antioxidant activity [35]. Due to the biological activity of gallic acid and its derivatives, it is widely used in food and pharmaceutical fields. Similarly, the anthers and pollen of tree peony contain gallic acid and its derivatives, such as gallic acid and methyl gallate, which also indicates their potential application value.

In terms of antibacterial activity, 13 and 55 metabolites were correlated with resistance to Gram-positive (*Staphylococcus aureus*, *Streptococcus hemolytis-β*, *Pseudomonas aeruginosa*, *Listeria monocytogenes*) and Gram-negative bacteria (*Pseudomonas aeruginosa*, *Escherichia coli*, *Proteus vulgaris*, *Salmonella enterica* subsp*. Enterica*), respectively. In addition, 11 metabolites were associated with all antibacterial activities, of which gallic acid and ferulic acid have a strong correlation with all antibacterial activities. Gallic acid and ferulic acid belong to phenolic compounds, which are one of the most abundant groups of substances in the plant kingdom. In previous studies, ferulic acid and gallic acid have shown relatively high antibacterial activities [36].

In enzyme inhibition activity, 54 metabolites are correlated with inhibition of α-glucosidase activity and 52 metabolites are associated with inhibition of α-amylase activity among all identified metabolites. Among them, gallic acid, isorhamnetin 3-robinobioside, paeoniflorin are significantly correlated with α-amylase inhibitory activity. Meanwhile, gallic acid hexoside, digallic acid, and pentagalloylglucose are significantly correlated with the inhibitory activity of α-glucosidase. Studies have shown that gallic acid has inhibitory activity on amylase, and in order to reduce the side effects of acarbose, gallic acid is also used in the treatment of diabetes [37]. Likewise, pollen was relatively low in gallic acid in all samples, possibly contributing to its weaker inhibitory activity on α-amylase.

#### 2.7.2. In Silico PASS Software Used for Predicting Biological Activity

DAMs, such as pentagalloylglucose, tetra-o-galloyl-beta-d-glucose, gallic acid, methyl gallate, gallic acid hexoside are present at higher levels in all samples by untargeted metabolomic assay. The current research focuses on the above-mentioned metabolites to speculate on the health benefits associated with them. Previous studies have demonstrated the feasibility of PASS 9.1 software for biological activity prediction [38]. Therefore, this computer algorithm was used to predict characteristic metabolite biological activities in *P. ostii*. The structure of metabolites was uploaded in Mol file format. By clicking the predict button, a table including (Pa) probable activity and Pi (probable inactivity) was obtained, and Pa > 0.7 was set to further obtain the maximum potential biological activity that can be used in future experimental models. The final prediction results showed that pentagalloylglucose had a high Pa value (Pa > 0.7) in free radical scavenger and anti-inflammatory. Similarly, tetra-o-galloyl-beta-d-glucose was detected with high Pa (Pa > 0.7) for free radical scavenger and antioxidant. In addition, gallic acid, gallic acid hexoside and methyl gallate were detected to have higher Pa in inhibiting amylase and glucosidase activities (Pa > 0.7) (Appendix A). Numerous studies have also shown that the active ingredients in tree peony have bioactive effects, which also provides support for the prediction [39].

## 3. Materials and Methods

### 3.1. Plant Materials

The anthers and pollen of *P. ostii* were collected from the *Paeonia* germplasm nursery of Northwest A&F University (34°15′ N, 108°03′ E, and Alt. 448 m) in 2022. *P. ostii* anthers at four different developmental stages (corresponding to the floral bud stage, initial bloom stage, full bloom stage and senescence stage) were collected and named S1, S2, S3, and S4, respectively; mature pollen was collected and named P (Figure 12). All samples were placed in 50 mL centrifuge tubes, freeze-dried, and stored at −80 °C for future analysis.

### 3.2. Reagents and Chemicals

Gallic acid, rutin, toluidine blue, DNS reagent, cephalexin, and ampicillin were purchased from Yuanye Biotechnology Co. (Shanghai, China). Acarbose, α-amylase, α-glucosidase, p-nitrophenyl-β-D-galactopyranoside (PNPG), 2,3,5-triphenyl tetrazolium chloride (TTC), 6-Hydroxy-2,5,7,8-tetramethylchromane-2-carboxylic acid (Trolox), Folin–Ciocalteu reagent, dimethyl sulfoxide (DMSO), 1,1-Diphenyl-2-picrylhydrazyl (DPPH), 2,2′-Azinobis-(3-ethylbenzothiazoline-6-sulphonate) (ABTS), and 2,4,6-tris(2-pyridyl)-s-triazine (TPTZ) were purchased from Sigma-Aldrich (St. Louis, MO, USA). Sodium hydroxide, hydrochloric acid, glacial acetic acid, ferric chloride, anhydrous sodium carbonate, potassium persulfate, sodium acetate, aluminum nitrate, sodium nitrite, and other regents were obtained from Bodi Chemical Co. (Tianjin, China). HPLC-grade formic acid, acetonitrile, and methanol were bought from CNW Technologies GmbH (Duesseldorf, Germany).

### 3.3. Paraffin Section

Paraffin sections were performed according to the following protocols as previously reported [40]. Briefly, anthers at different stages were fixed, dehydrated, transparent, embedded, sectioned, deparaffinized, stained with 1% toluidine blue solution for 10 min, washed with distilled water, and mounted with neutral gum, and then placed under an optical microscope (Olympus BX43) for observation.

### 3.4. Sample Pretreatment

Approximately 1 g of sample powder was mixed with 20 mL of methanol, and ultrasonically extracted at room temperature for 30 min, followed by centrifugation at 12,000 rpm for 10 min at 4 °C and collect the supernatant. The extraction process was repeated three times, then the supernatants were pooled together, concentrated under reduced pressure, freeze-dried, and then dissolved in methanol or DMSO solution for future analysis. HPLC-grade methanol sample solutions were filtered through a 0.22 μm membrane before UPLC-QTOF-MS analysis, quality control (QC) samples were the equal volume mixtures of methanol extracts of each sample, and all samples were stored at −80 °C for following analyses.

### 3.5. Total Phenolic Content (TPC) Assay

TPC was measured using the reported protocol [41]. Volumes of 0.4 mL of deionized water, 0.1 mL of 1 M Folin–Ciocalteu reagent and 0.1 mL sample extract solution were mixed, and incubated at room temperature in the dark for 5 min. Then, the 0.4 mL Na_2_CO_3_ solution (7.5%, *w*/*v*) was added to the mixture and incubated in the dark for 2 h. Finally, the mixture was added to a 96-well plate, and the absorbance was measured at 760 nm using a microplate reader (SP-Max 2300A2, Shanghai, China). TPC was expressed as milligrams of gallic acid equivalents per gram of sample in dry weight (mg GE/g DW).

### 3.6. Total Flavonoid Content (TFC) Assay

TFC was measured using the previously reported method [42]. A volume of 0.15 mL NaNO_2_ (5%, *w*/*v*) solution, 0.15 mL Al(NO_3_)_3_ (10%, *w*/*v*) solution and 0.5 mL sample extract solution were mixed. After 6 min incubation at room temperature, add 2 mL NaOH (4%, *w*/*v*) solution and incubated for 3 min. Finally, the mixture was added to a 96-well plate, and the absorbance was measured at 510 nm using a microplate reader. TFC was expressed as milligrams of rutin equivalents per gram of sample in dry weight (mg RE/g DW).

### 3.7. Antioxidant Activity Assay

#### 3.7.1. DPPH^•^ Scavenging Assay

DPPH free radical scavenging assay is a HAT reaction widely used to determine the antioxidant activity [21]. The DPPH^•^ scavenging ability was determined based on an improved method. Briefly, equal volumes of sample extract solution and DPPH^•^ solution (1 mM) were mixed. After 45 min of reaction in the dark, the absorbance was measured at 517 nm using a microplate reader. The antioxidant activity of the samples was expressed as milligrams of Trolox equivalents per gram of sample in dry weight (mg TE/g DW).

#### 3.7.2. ABTS^+•^ Scavenging Assay

ABTS free radical scavenging assay involves both hydrogen atom transfer and single electron transfer and is widely used to measure the antioxidant activity of various plant foods containing hydrophilic, lipophilic and highly pigmented antioxidant compounds [43]. The fresh ABTS^+•^ working solution was prepared based on previous description [44]. Equal volumes of sample extract solution and ABTS^+•^ working solution were mixed. After incubating for 10 min, the absorbance was measured at 734 nm using a microplate reader. The antioxidant activity result of samples was expressed as milligrams of Trolox equivalents per gram of sample in dry weight (mg TE/g DW).

#### 3.7.3. FRAP Assay

The FRAP assay is a total electron transfer (ET) reaction that is often used in conjunction with other methods to distinguish the main mechanism of different antioxidants [45]. FRAP working solution was prepared with reference to previous studies and kept at 37 °C until analysis [46]. A volume of 50 μL sample extract solution was mixed with 900 μL FRAP working solution, followed by a 10 min reaction. After that, the absorbance was measured at 593 nm using a microplate reader. The antioxidant result of sample was expressed as milligrams of Trolox equivalents per gram of sample in dry weight (mg TE/g DW).

### 3.8. Antibacterial Activity Assay

Four Gram-positive bacteria (*Staphylococcus aureus*, *Streptococcus hemolytis*-*β*, *Propionibacterium acnes*, and *Listeria monocytogenes*) and four Gram-negative bacteria (*Pseudomonas aeruginosa*, *Escherichia coli*, *Proteus vulgaris*, and *Salmonella enterica* subsp. *enterica*) were used in the antibacterial assay. The minimum inhibitory concentration (MIC) method was performed as previously reported [47], with some modifications. Streptococcus hemolytis-β was cultured in tryptone soy broth (TSB) and other bacteria were cultured in nutrient agar. A series of 2% DMSO-PBS sample solutions were prepared by the double dilution method. In each well of a 96-well plate, 100 μL of bacterial suspension (1 × 10^7^ CFU/mL) and 100 μL of sample extract solutions of different concentrations were mixed, then 10 μL of TTC solution (5 mg/mL) was added, and the plate containing the bacterial solution was incubated at 37 °C for 24 h. Cefalexin and ampicillin were used as positive controls.

### 3.9. Enzyme Inhibition Assay

#### 3.9.1. α-Glucosidase Inhibition Assay

α-Glucosidase inhibition activity was detected according to following protocols as previously reported [48], with slight modifications. In each well of a 96-well plate, different concentrations of samples or acarbose (50 μL) were mixed with α-glucosidase (20 μL, 0.2 U/mL, dissolved in 0.1 mM, pH 6.9 PBS) and pNPG solution (20 μL, 2.5 mM). After incubation at 37 °C for 15 min, Na_2_CO_3_ (100 μL, 0.2 M) solution was added to terminate the reaction. The absorbance was measured at 405 nm using a microplate reader. Acarbose as the positive control.

#### 3.9.2. α-Amylase Inhibition Assay

α-Amylase inhibition activity was analyzed according to previously reported protocols [49], with slight modifications. In brief, samples or acarbose (10 µL) of different concentrations were mixed with α-amylase (10 μL, 0.5 U/mL, dissolved in 0.1 mM, pH 6.9 PBS) at 37 °C for 15 min, followed by the addition of starch solution (500 μL, 1%). After incubation at 25 °C for 10 min, add DNS reagent (500 μL) and put the solution into boiling water 10 min to terminate the reaction. After cooling, distilled water was added to make up to 5 mL. The absorbance at 540 nm was measured using microplate reader. Acarbose as the positive control.

### 3.10. Untargeted Metabolomics Analysis

#### 3.10.1. UPLC-QTOF-MS Analysis

UPLC analysis was performed on a Shimadzu LC-30A (Shimadzu Corp., Kyoto, Japan). The chromatographic separation was conducted with an ACQUITY UPLC HSS T3 Column (100 mm × 2.1 mm, 1.8 µm; Waters, MA, USA). The mobile phase consisted of 0.1% formic acid in water (A) and acetonitrile (B), with a flow rate of 0.3 mL/min. The injection volume of each sample was 1 µL and column temperature was set to 35 °C. The gradient elution procedure was based on previous report [50].

Mass spectrum acquisition experiments were performed using an AB SCIEX Triple TOF 5600 plus mass spectrometer system coupled to an electrospray ionization (ESI) source (AB SCIEX, Foster City, CA, USA). The metabolome data of each sample, including QC, were acquired using information-dependent acquisition (IDA) with full-mass scan acquisition in the negative ion mode, with spectrum acquisition ranging from 50 to 1500 m/z. Other MS parameters were set as the published reports [19].

#### 3.10.2. Data Processing

MS-DIAL is an open-source mass spectrometry data-independent analysis software, was used for data processing [51]. Before processing date with MS-DIAL, the raw data files obtained from the AB SCIEX system were converted to Analysis Base File (ABF) format through Analysis Base File Converter software (version 1.1) for format compatibility. ABF format file was imported into MS-DIAL 4.38 software to extract retention time, peak height, peak area, m/z value, addition of the combined ions, signal-to-noise ratio (S/N) of samples. Accurate mass tolerances for metabolite identification were set to 0.01 and 0.025 Da for MS1 and MS2, respectively. For adduct ion selection, [M−H]^−^, [M−2H]^−^ and [M+HCOO]^−^ were selected in the negative ion mode. The QC samples were used as peak alignment files with retention time tolerance of 0.05 min and a MS1 tolerance of 0.015 Da. Data normalization was performed with total ion chromatograms to eliminate differences between samples for comparison between samples. Finally, metabolite identification was performed using public databases, such as Massbank (https://massbank.eu/MassBank/), MoNa (https://mona.fiehnlab.ucdavis.edu/), ReSpect (http://spectra.psc.riken.jp/), GNPS (https://gnps.ucsd.edu/ProteoSAFe/static/gnpssplash.jsp), and MS-DIAL (http://prime.psc.riken.jp/compms/msdial/main.html#MSP), all databases are accessed on 21 October 2023.

### 3.11. Multivariate Statistical Analysis

All measurements were performed in triplicate. The UPLC-QTOF-MS raw data were uploaded to Metaboanalyst 5.0 (https://www.metaboanalyst.ca/) for data filtering. The clean data was used for PCA analysis. Then, all normalized data were exported for subsequent bioinformatics analysis [52]. PLS-DA was carried out using SIMCA 14.1 software (Umetrics, Malmo, Sweden). Heatmaps were generated using the pheatmap package in R 4.2.1 to reflect the relative contents of different metabolites in different samples. Volcano maps, line chart and histogram were generated using the ggplot2 package in R. The correlation between samples was analyzed and visualized using the corrplot package in R. The pathway enrichment analysis was conducted using Kyoto Encyclopedia of Genes and Genomes (KEGG) (http://www.genome.jp/kegg/) and MetaboAnalyst databases, and pathway diagrams were generated using the Pathview package in R.

## 4. Conclusions

This study systematically compared the phytochemicals of *P. ostii* anthers using the untargeted metabolomics approach by UPLC-QTOF-MS. A total of 105 metabolites were initially identified. At the superclass level, these metabolites mainly belong to phenylpropanoids and polyketides, organic acids and derivatives, benzenoids, lipids and lipid-like molecules, and organic oxygen compounds. Among the DAMs, carbohydrates and flavonoids were the predominant metabolites in *P. ostii* pollen, while pentagalloylglucose, tetra-o-galloyl-β-d-glucose, gallic acid and methyl gallate were the predominant metabolites in anthers. In the comparison groups, there were more DAMs in S4 vs. P, which were mainly enriched in “starch and sucrose metabolism”, “flavonoid biosynthesis” and “galactose metabolism”. Biological activity evaluation showed that *P. ostii* anthers and pollen both have antioxidant activity, antibacterial activity, and inhibitory activity of α-amylase and α-glucosidase, the anther stage of S4 shows the highest biological activity generally. PASS biological prediction analysis detected higher predictive possibility of antioxidant capacity and anti-inflammatory of pentagalloylglucose and tetra-o-galloyl-beta-d-glucose, while gallic acid, gallic acid hexoside and methyl gallate were predicted to have inhibitory activity to α-amylase and α-glucosidase. In conclusion, this study provides insights into the metabolism of *P. ostii* anther and pollen, which have potential value for development into functional foods and medicines.

## Figures and Tables

**Figure 1 ijms-25-05462-f001:**
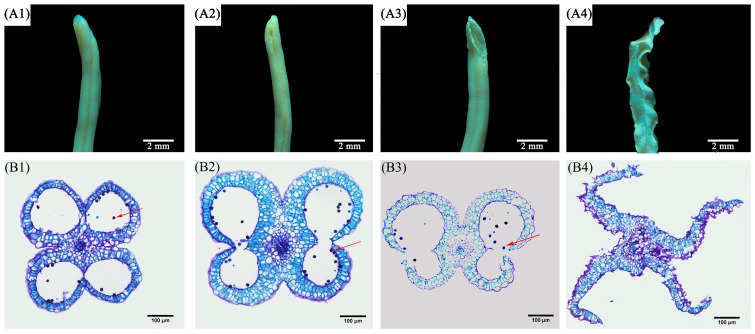
Microscopic observation ((**A1**–**A4**): S1–S4, Bar = 2 mm) and paraffin section of anthers at different developmental stages ((**B1**–**B4**): S1–S4, Bar = 100 μm). Red arrows indicate pollen grains.

**Figure 2 ijms-25-05462-f002:**
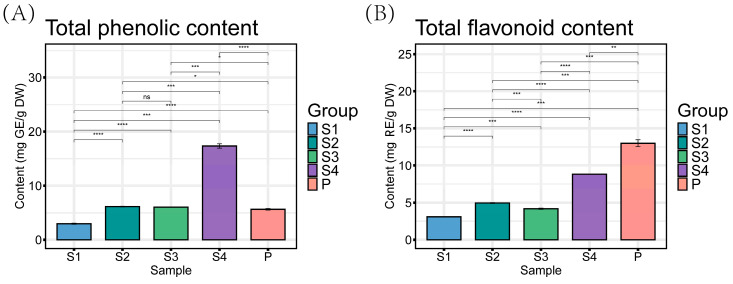
The contents of total phenolic (**A**) and total flavonoid (**B**) of anthers (S1, S2, S3, and S4) and pollen (P). Values are the means ± standard deviations, *n* = 3. Note: * corresponds the significant difference of the *t*-test * *p* < 0.05, ** *p* < 0.01, *** *p* < 0.001, **** *p* < 0.0001, ns indicates there is no significant difference.

**Figure 3 ijms-25-05462-f003:**
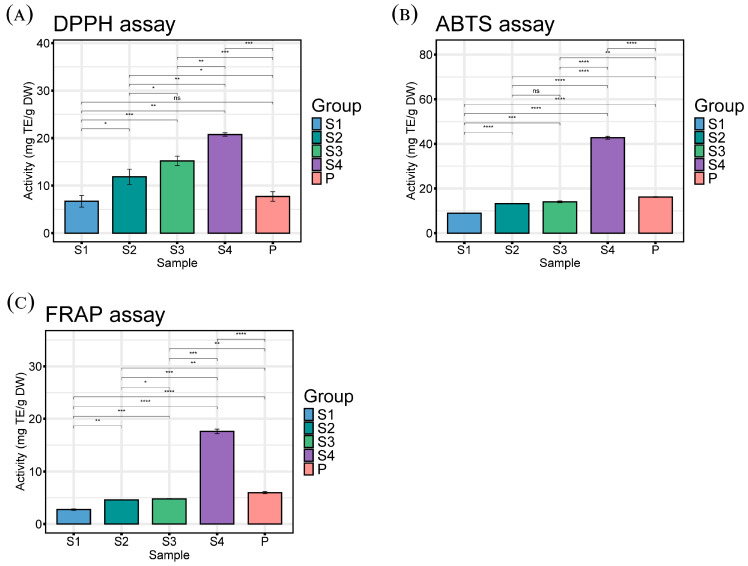
The results of DPPH assay (**A**), ABTS assay (**B**), FRAP assay (**C**) of anthers (S1, S2, S3, and S4) and pollen (P). Values are the means ± standard deviations, *n* = 3. Note: * corresponds the significant difference of the *t*-test * *p* < 0.05, ** *p* < 0.01, *** *p* < 0.001, **** *p* < 0.0001, ns indicates there is no significant difference.

**Figure 4 ijms-25-05462-f004:**
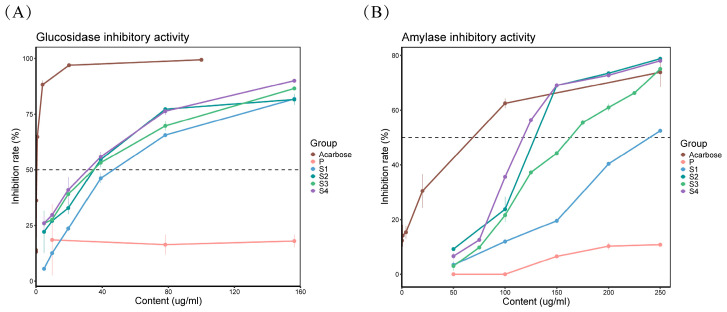
The results of α-glucosidase (**A**) and α-amylase (**B**) inhibitory activity of anthers (S1, S2, S3, and S4) and pollen (P), with acarbose as the positive control.

**Figure 5 ijms-25-05462-f005:**
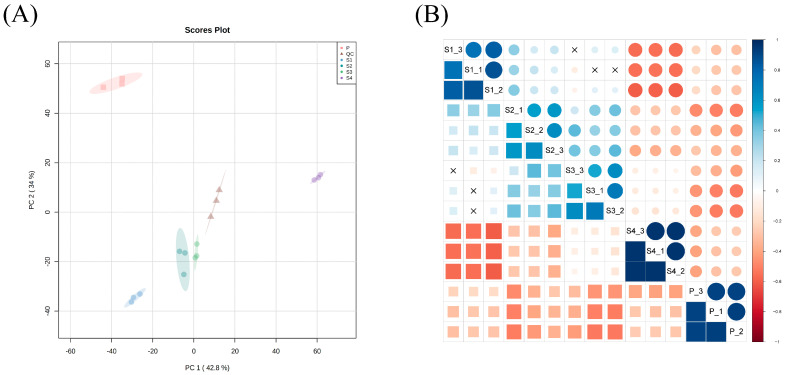
Data characterization of all detected ion peaks of anthers (S1, S2, S3, and S4) and pollen (P) based on untargeted metabolomics. (**A**) The PCA score scatter plot; (**B**) correlation analysis between samples; different colors indicate differences in correlation between samples, blue indicates positive correlation, and red indicates negative correlation.

**Figure 6 ijms-25-05462-f006:**
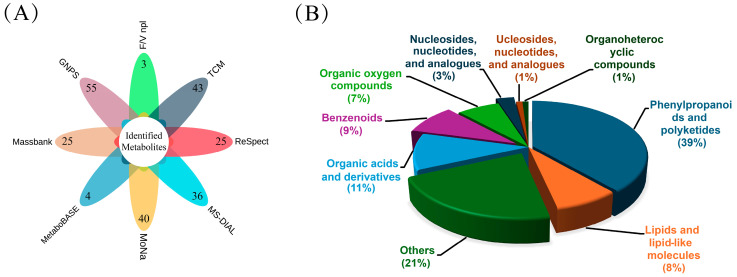
Metabolites identification and classification. (**A**) Number of metabolites identified by each database; (**B**) compositional analysis of all identified metabolites at the superclass level.

**Figure 7 ijms-25-05462-f007:**
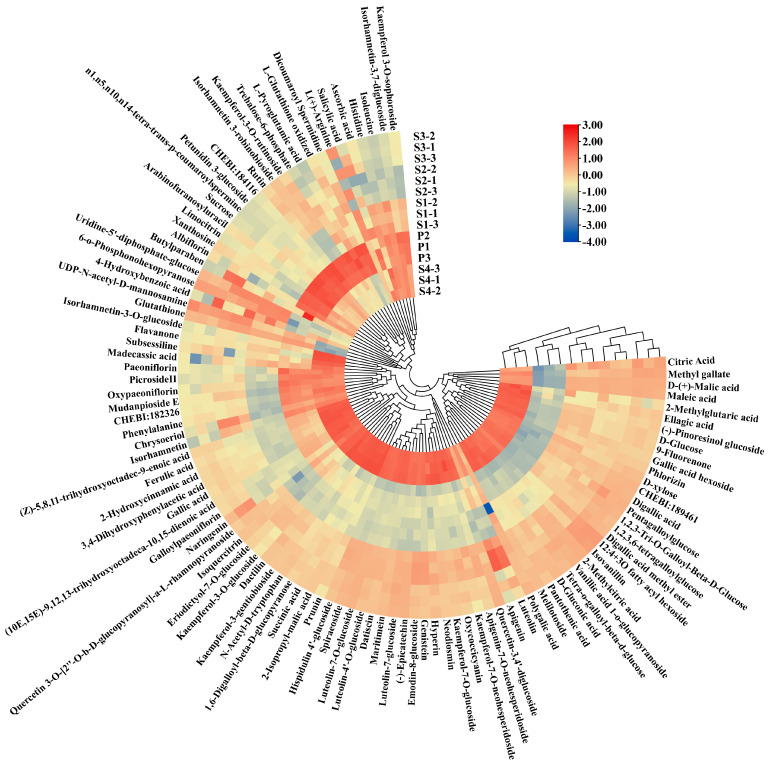
Hierarchical analysis (HCA) heat map of all identified metabolites in different samples. Red and blue indicate higher and lower abundances, respectively.

**Figure 8 ijms-25-05462-f008:**
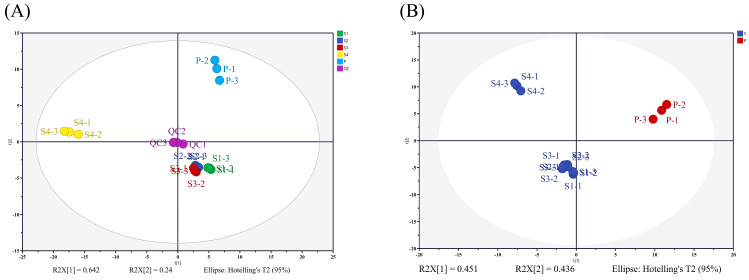
Multivariate statistical analysis of samples. (**A**) The PCA score scatter plot of different samples and QC; (**B**) the PLS-DA score scatter plot of different samples.

**Figure 9 ijms-25-05462-f009:**
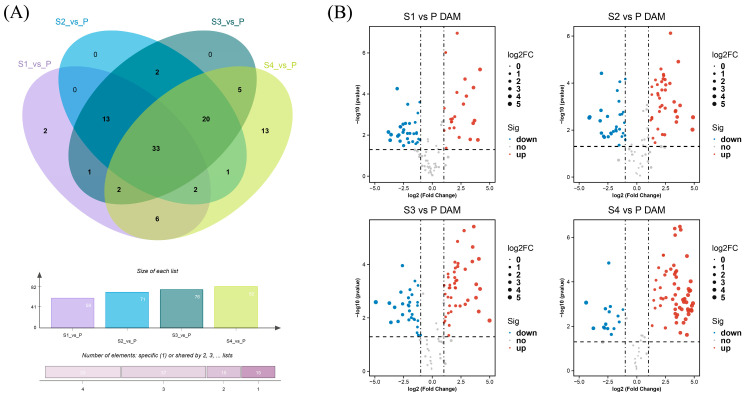
Differentially accumulated metabolites (DAMs) between different samples. (**A**) Venn diagram of different comparison groups; (**B**) volcano plot of different comparison groups.

**Figure 10 ijms-25-05462-f010:**
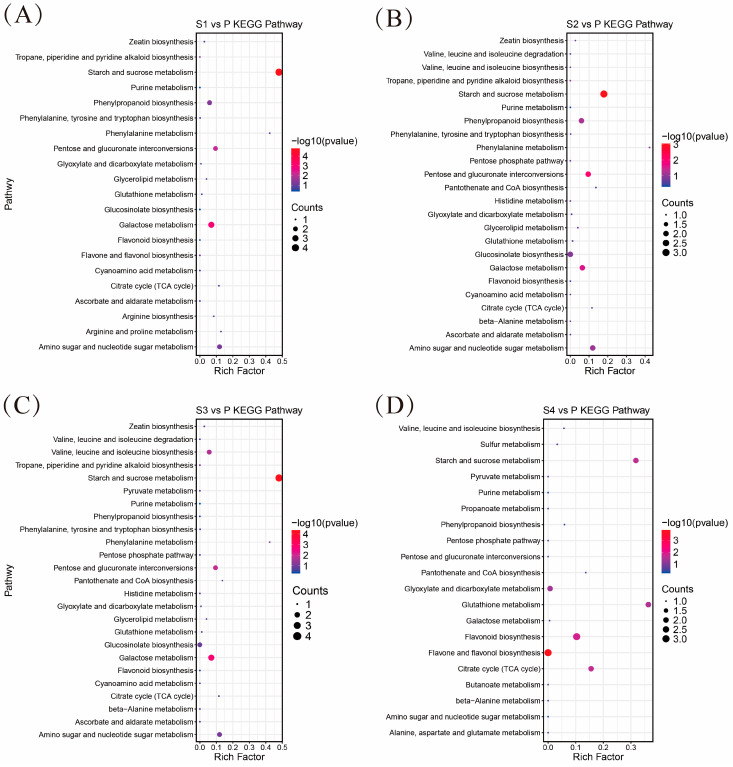
KEGG pathways of differentially accumulated metabolites (DAMs) identified among comparisons. (**A**) KEGG pathway analysis of S1 vs. P; (**B**) KEGG pathway analysis of S2 vs. P; (**C**) KEGG pathway analysis of S3 vs. P; (**D**) KEGG pathway analysis of S4 vs. P.

**Figure 11 ijms-25-05462-f011:**
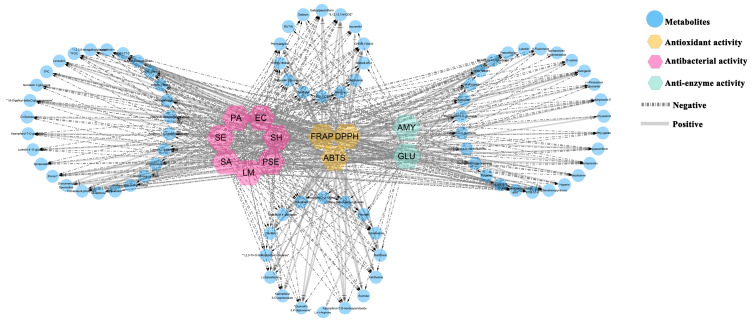
Pearson correlation network (|r| > 0.5, *p* ≤ 0.05) between metabolites and biological activities. AMY: α-amylase; GLU: α-glucosidase; SA: *Staphylococcus aureus*; EC: *Escherichia coli*; SE: *Salmonella enterica* subsp*. Enterica*; SH: *Streptococcus hemolytis-β*; LM: *Listeria monocytogenes*; PA: *Pseudomonas aeruginosa*; PSE: *Pseudomonas aeruginosa*.

**Figure 12 ijms-25-05462-f012:**
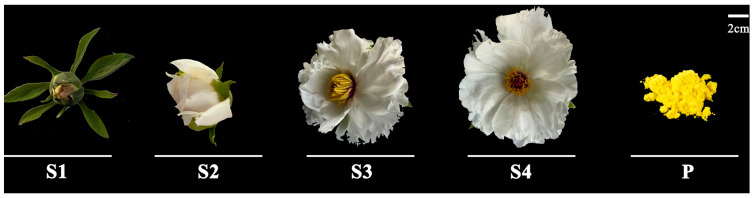
Phenotypic variation in different samples. Bar = 2 cm.

**Table 1 ijms-25-05462-t001:** The minimum inhibitory concentration (mg/mL) of anthers and pollen against test bacteria.

Sample	Minimum Inhibitory Concentration (mg/mL)							
	*Staphylococcus aureus*	*Streptococcus hemolytis-β*	*Propionibacterium acnes*	*Listeria monocytogenes*	*Pseudomonas aeruginosa*	*Escherichia coli*	*Salmonella enterica* subsp. *Enterica*	*Proteus vulgaris*
S1	100	25	100	25	100	100	100	100
S2	100	25	100	25	100	100	100	100
S3	50	12.5	50	25	100	100	100	100
S4	25	12.5	50	12.5	25	50	50	100
P	100	25	100	6.25	12.5	6.25	50	100
Cef	0.25	1	0.5	0.125	0.25	0.25	0.125	0.25
Amp	0.0625	0.5	1	0.0625	0.125	0.125	0.0625	0.25

## Data Availability

All relevant data are included in the manuscript and Appendix A.

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
