# Peer review of "Metabolite Profiling and Biological Activity Assessment of Paeonia ostii Anthers and Pollen Using UPLC-QTOF-MS"

_ijms, 2024, doi:10.3390/ijms25105462_

Round 1

Reviewer 1 Report

Comments and Suggestions for Authors

Review comment can be found in the file attached.

Author Response

Thank you for your positive feedback and valuable suggestions on this manuscript.It is true that there are deficiencies in the formatting of this manuscript, and I will follow your comments to make improvements on a case-by-case basis, so that the manuscript will be more standardised! (Detailed response are attached as PDF)

Reviewer 2 Report

Comments and Suggestions for Authors

The manuscript proposed by the authors is based on the methodology of non-targeted metabolomics. This is a relatively new and fashionable direction in plant biochemistry, with the noble aim of describing the pool of products of primary and secondary metabolism, as well as the intermediary metabolites of their biosynthesis, in as complete and detailed a manner as possible.

 The authors attempted to apply a similar technique to the characterisation of Paeonia pollen and anthers in order to identify potentially useful compounds with high biological activity.

Having carefully studied the manuscript, especially the methodological part, I regret to admit that the results obtained cannot be called a complete characterisation of the metabolome, as the authors claim in the introduction.

Firstly, the authors used a simple ultrasonic methanol extraction followed by filtration for the UPLC-TOF-MS analysis. This step already indicates that in this case it is not possible to characterise the metabolome in the strict sense, since the extract will only contain metabolites that are soluble in a very polar solvent (methanol). Such compounds represent at best a third of the pool of molecules present in plant tissues and cells. The authors could have achieved much better results by using two-phase extraction according to the protocols of Folch or Bligh & Dyer, etc., with sequential analysis of the upper and lower phases of the extract.

Secondly, even the molecules identified by the authors (assuming that relying only on the exact mass of the molecular ion and one MRM transition is sufficient for accurate identification using open databases) cannot be considered fully native, since the authors did not use any antioxidants during extraction, which prevents oxidation of, for example, polyunsaturated fatty acids, carotenoids and even lipids.

It is significant, for example, that the authors identified free fatty acids and only lyso forms of phospholipids. It is well known that fatty acids are never present in free form in plant cells, as they have a damaging effect on membranes, which in turn cannot consist exclusively of lysophospholipids.

The discovery of only such metabolites associated with the lipidome indicates a gross methodological error, both in terms of extraction, sample preparation and identification, which can also be assessed by another marker. The fact is that QTOF-ESI-MS and even MS2 do not allow to determine the exact position of the ethylene bond in the fatty acyl, let alone its geometry, so the trans-9-palmitoelaidic acid identified by the authors is also an artefact of the methodology.
I will not bother to list all these c
omments.

Thirdly, each group of compounds requires its own approach, not only in terms of extraction, derivatisation and conditions for HPLC analysis (different stationary and mobile phases and detection methods), but even different conditions for recording mass spectra.

 For example, lipids from different classes may require different collision energies to obtain a second order mass spectrum sufficient for identification. Other groups of molecules should also be approached with caution.

 The authors' data should be considered preliminary, regardless of these criticisms. However, the authors' identification of several groups of metabolites through their pool of bioinformatics analyses is promising and justifies further work on their accurate identification using correct methods. It is unfortunate that such work was not carried out, but it is important to approach this task with confidence and a clear plan of action. To meet the publication standards of the proposed journal, the manuscript must contain a more thorough discussion of the obtained data and theoretical generalizations that demonstrate scientific novelty.

Reviewer 3 Report

Comments and Suggestions for Authors

The article "Metabolite Profiling and Biological Activity Assessment of Paeonia ostii Anthers and Pollen Using UPLC-QTOF-MS" provides a comprehensive analysis of the metabolite profiles and biological activities of Paeonia ostii anthers and pollen. The article makes a significant contribution to the understanding of the metabolite profiles and biological activities of Paeonia ostii anthers and pollen. Its comprehensive methodology, detailed analysis, and clear presentation of results are notable strengths. Some of my observations are:

·         While the article provides an in-depth analysis of Paeonia ostii, a comparative analysis with other Paeonia species or related plants is needed to offer a better context to the results.

·         The article could benefit from a more detailed discussion on the potential mechanisms of action for the observed biological activities.

·         The study could further address potential variability due to environmental factors, cultivation practices, and sample preparation methods. Discussing how these factors might affect the metabolite profiles and biological activities would enhance the robustness of the findings.

Author Response

Thank you for your valuable suggestion. I agree that my discussion can be further supplemented to make the content of the article more detailed. I will improve my manuscript according to your suggestions. Thank you for your comments

Reviewer 4 Report

Comments and Suggestions for Authors

The reported study explores the metabolite profiles and biological activities of Paeonia ostii, an economically significant oil crop known for its health benefits. The authors identified 129 metabolites in anthers and pollen, revealing differences in composition. Both extracts exhibited antioxidant, antibacterial, and enzyme inhibitory activities, with the S4 anther stage showing the highest biological activity. This research enhances understanding of P. ostii and suggests potential avenues for its further utilization.

I am extremely satisfied with the manuscript provided. The methods and procedures outlined are impeccably detailed and easily repeatable, instilling confidence in the reliability of the research. The authors conducted extensive analyses, and the information obtained correlates seamlessly, demonstrating a thorough and systematic approach to the study. Overall, I commend the authors for their meticulous work, which has contributed significantly to our understanding of the subject matter.

Notes and recommendations:

Row 55 – space between the references 15, 16

The references are not in the journal style. Please make them comply with the style of the journal.

The aim of the study is briefly outlined in lines 66 and 67. In light of this, I would kindly request the authors to provide a more detailed description of the potential applications and uses of the analyzed objects. Please insert it in the introduction.

The findings from in silico studies provide additional support to the experimental results, but they are not very well interpreted.

The in vitro biological studies serve to strengthen the experimental findings, albeit with a lack of thorough interpretation. Typically, in silico results are employed to bolster experimental data. In this context, a comparison between the biological test results and those obtained in silico could be made, followed by commentary. However, in the accompanying file, aside from Gallic acid hexoside and Tetra-o-galloyl-beta-d-glucose, no predicted antioxidant properties were observed. Please comment on those results.

I'm curious about the authors' choice of methods for assessing antioxidant activity. The selected methods seem to focus on measuring total radical activity, which could be influenced by the solvents used, potentially quenching harmful radicals. Wouldn't it be more appropriate to utilize methods that evaluate the antioxidant properties of the samples against real reactive oxygen species (ROS) generated in the human body? There are several in vitro methods available for such purposes.

I have a generally positive impression of the article, and therefore, I recommend its publication after minor revisions.

Round 2

Reviewer 2 Report

Comments and Suggestions for Authors

I would like to begin by thanking the authors for providing a comprehensive list of literature references in response to the comments made in the initial review. However, it is beyond the scope of my review to assess these papers as part of my assessment of the authors' manuscript. Furthermore, I do not consider the argument that "this method has been used by many teams that have published N articles" to be a sufficient counterargument to the substantive comments made in the manuscript in question.

Having carefully considered the responses to the review and the revised manuscript, I would like to make the following points.

Firstly, I cannot agree with the authors' assertion that "before the experiment, we could not predict all the metabolites contained in peony anthers". It is evident that the tissues of almost any plant are composed of over 90 per cent water. The remaining percentage of the dry weight will be comprised of proteins, lipids (neutral and polar), the majority of which are esterified fatty acids; carbohydrates in the form of monosaccharides, disaccharides and polysaccharides; and no more than 0.5% of the dry weight will be derived from secondary metabolite compounds, including flavonoids, anthocyanins, carotenoids, isoprenoids, etc.

The proportions of these classes of substances can vary depending on the function of the plant's tissues or organs. It is evident that non-target metabolomics is not necessary to reach this conclusion. Ultimately, the authors reached similar conclusions, as can be seen by the line "Multivariate statistical analysis revealed the metabolite differences between anthers and pollen, with higher lipid content in pollen and higher phenolic content in anthers" in the Abstract of the manuscript.

Given the authors' conclusions are based on extensive research, it is imperative that further work is conducted to accurately and precisely identify the lipid and phenolic compounds present. Without these results, which will confirm and refine the identification data obtained through non-target metabolomics methods, the current work lacks credibility. This is due to the following reasons, which the authors did not address in their reply:

Firstly, it is unlikely that the polar lipids of higher plants will consist exclusively of lyso-forms and free fatty acids. Both are generally considered to be an artefact of storage, preservation, extraction or derivatisation techniques until proven otherwise. For further information, please refer to the following link:
https://lipidlibrary.aocs.org/lipid-analysis/selected-topics-in-the-analysis-of-lipids/preparation-of-lipid-extracts-tissues.

Secondly, UPLC-TOF-MS/MS methods are not sufficiently reliable for the identification of FAs, as neither the position of the double bond nor its geometry can be determined from the mass of the molecular ion. This method allows the number of ethylene bonds to be established, but this is insufficient for a complete judgement on the chemical structure of a fatty acid.

In light of these comments, the part of the article that operates on lipid and fatty acid data should either be removed from the manuscript or verified by adequate methods of targeted metabolomics. The manuscript can then be considered again for publication in the journal.

Author Response

Many thanks for your insightful comments on this revised manuscrip. Indeed, MS2 alone is not sufficient to accurately identify fatty acid, and thus this manuscrip may contain numerous inaccuracies in the identification of FAs. Therefore, I will modify or delete the related data in this part to make the results in this manuscrip more reliable.

Round 3

Reviewer 2 Report

Comments and Suggestions for Authors

Firstly, I would like to express my gratitude to the authors for their willingness to make changes to the manuscript in order to enhance the quality of the data presented and their validity. Following the removal of the most controversial results from the manuscript, it can now be characterised as meeting the quality standards of the journal.

Nevertheless, there are a number of minor comments that we would also like to draw the authors' attention to. 

In Table S1 (and also Tables S2-S4), which accompany the publication, there remain unidentified compounds that it would be advisable to remove. These are substances that have been labelled with codes that are not clear to either the reader or the reviewers. For example, 3083-77-0 or CHEBI:XXXXXXX and similar.

Rows containing such codes should either be deleted or the substance name should be edited in accordance with accepted nomenclature in biochemistry. This remark applies to all tables where such compounds occur. Subsequently, a similar revision should be made in the text of the article and in the illustrations. For example, Fig. 7 contains these ‘strange’ molecules.

Furthermore, I recommend that the mass values be brought to 4 decimal digits in all tables and columns. 

The total ion current (TIC), at a wavelength of 525 nanometres (nm) for anthocyanins, 275 nm for flavonoids, and 450 nm for carotenoids, should be arranged one beneath the other and aligned on a time scale. This is typically the case for high-performance liquid chromatographic-mass spectrometric (HPLC-MS) systems, which usually have a ultraviolet (UV) or diode array detector (DAD) matrix detector situated before the mass spectrometric detector. The substances identified above the peaks should then be labelled. These data will clearly reflect the efficiency of the separation process and will undoubtedly be useful to other researchers in their choice of working methods.
